# Understanding the Role of Organic Matter Cycling for the Spatio-Temporal Structure of PCBs in the North Sea

**Ute Daewel** [1,*], **Evgeniy V. Yakushev** [2,3,*], **Corinna Schrum** [1,4,5], **Luca Nizzetto** [2] and **Elena Mikheeva** [1]

[1]  Helmholtz-Zentrum Geesthacht, Institute of Coastal Research, Max-Planck-Str. 1, 2152 Geesthacht, Germany; corinna.schrum@hzg.de (C.S.); elena.mikheeva@hzg.de (E.M.)
[2]  Norwegian Institute for Water Research (NIVA), Gaustadalléen 21, 0349 Oslo, Norway; Luca.nizzetto@niva.no
[3]  Shirshov Institute of Oceanology, Russian Academy of Sciences, 36 Nakhimovskiy prospekt, 117991 Moscow, Russia
[4]  Institute of Oceanography, Universität Hamburg, Bundesstraße 53, 21046 Hamburg, Germany
[5]  Geophysical Institute, University of Bergen and Hjort Centre for Marine Ecosystem Dynamics, Allegaten 41, 5007 Bergen, Norway
*  Correspondence: ute.daewel@hzg.de (U.D.); evgeniy.yakushev@niva.no (E.V.Y.)

**Abstract:** Using the North Sea as a case scenario, a combined three-dimensional hydrodynamic-biogeochemical-pollutant model was applied for simulating the seasonal variability of the distribution of hydrophobic chemical pollutants in a marine water body. The model was designed in a nested framework including a hydrodynamic block (Hamburg Shelf Ocean Model (HAMSOM)), a biogeochemical block (Oxygen Depletion Model (OxyDep)), and a pollutant-partitioning block (PolPar). Pollutants can be (1) transported via advection and turbulent diffusion, (2) get absorbed and released by a dynamic pool of particulate and dissolved organic matter, and (3) get degraded. Our model results indicate that the seasonality of biogeochemical processes, including production, sinking, and decay, favors the development of hot spots with particular high pollutant concentrations in intermediate waters of biologically highly active regions and seasons, and it potentially increases the exposure of feeding fish to these pollutants. In winter, however, thermal convection homogenizes the water column and destroys the vertical stratification of the pollutant. A significant fraction of the previously exported pollutants is then returned to the water surface and becomes available for exchange with the atmosphere, potentially turning the ocean into a secondary source for pollutants. Moreover, we could show that desorption from aging organic material in the upper aphotic zone is expected to retard pollutants transfer and burial into sediments; thus, it is considerably limiting the effectiveness of the biological pump for pollutant exports.

**Keywords:** PCBs; modeling; biological pump; North Sea; POPs

---

## 1. Introduction

The ocean plays a vital role in controlling the environmental transport and overall fate of many chemical pollutants that are relevant to the global environment and human population. These include, among others, persistent organic pollutants (POPs), including polyaromatic hydrocarbons (PAHs). Polychlorinated biphenyls PCBs are highly hydrophobic and persistent pollutants with a high tendency to bioaccumulate into biota, especially in organs or structures with high lipid contents, such as cell membranes and fatty tissues. PCB compounds are toxic at low concentrations to animals and humans,

causing reproductive and development problems as well as harming the immune system following chronic exposure [1,2]. These contaminants have long half-lives and can be especially detrimental in pristine areas due to long-range transport in the atmosphere and ocean [3]. These pollutants were mainly produced in the last century and mainly used in electrical equipment (including electrical capacitors, electrical transformers, and vacuum pumps) [4,5]. Though the production of these compounds is nowadays forbidden, there are still emitting primary sources such as poorly managed old stockpiles and PCB-containing devices that are still in use or inadequately dismissed [5], as well as secondary sources such as contaminated soils and sediments that are actively releasing these contaminants into air and water. Despite the relatively low concentrations that have been measured in marine waters, the presence of PCBs in marine systems is of concern. They can, in fact, biomagnify, increasing concentration by several orders of magnitude in the tissues of marine organisms at the higher trophic levels, also including populations of fish and other marine resources that are relevant for fisheries and human consumption. The impacts of PCBs on the exploitability and safety of these marine resources has been debated in several previous assessments and policy documents, e.g., [6–9].

Marine biogeochemical processes are relevant for controlling the environmental fate of marine chemicals on both global and regional scales [10–13]. In particular, the biological pump has been identified as a key driving process for controlling one of the major environmental sinks for hydrophobic POPs [11,12]. POPs reach the marine system from deposition and diffusive exchange with the atmosphere. In the coastal zone, they are mainly introduced to the marine system by discharges from land. Once in the euphotic zone, they undergo multimedia partitioning, which favors the particulate and dissolved organic phases. Due to the high hydrophobicity (measured through the octanol–water equilibrium partitioning coefficient ($K_{ow}$)), uptake by organic carbon (OC) pools, such as primary producer biomass, can influence chemical concentrations in the abiotic medium (namely water and dissolved organic carbon), thereby affecting the magnitude of advection [14,15]. The dependence of PCBs' multimedia distribution and transport fluxes on primary production in aquatic systems has been empirically observed in some studies [15–17].

In productive systems, such as the North Sea (Figure 1), the pool of more hydrophobic POPs that are adsorbed onto phytoplankton potentially represent the largest fraction of the total POP budget in surface water [14,18]. This pool is subsequently converted to detritus by phytoplankton aging and zooplankton grazing and digestion, and it is exported to deeper waters by the settling of detritus. It has been shown that the "biological pump" can deplete the concentration of dissolved hydrophobic, persistent chemicals in the surface water and, thus enhance net gaseous deposition from the atmosphere, thus acting as a major sink for both airborne and waterborne POPs [11,14,19]. Experimental evidence has confirmed this aspect [12]. Since this process occurs at large spatial scales, it was previously suggested that the biological pump might affect the transport and global fractionation of atmospheric POPs [20], as well as the general global scale mass balance of these chemicals.

Though this conceptual framework is currently broadly accepted, it does not take into consideration the reversibility of the binding between POPs and organic material (OM), and, in particular, it does not include (even at a very basic level) the role of the transformation to which OM is exposed to in the course of its synthesis, decay, and sinking. Since about 98% of the global oceanic primary production is remineralized in the mesopelagic zone, it is important to take these processes into account. Since the degradation half-lives of many POPs in water [21] are much longer than those of OM, we can hypothesize that there may be important rearrangements of POPs' vertical distributions among phases at this level. The decoupling between settling material and POP fluxes may hamper the effectiveness of the biological pump and lead to a temporary accumulation of chemicals in the dissolved phases in the mesopelagic zone. A fraction of this pool may finally re-diffuse toward surface waters and return to environmental cycling (e.g., re-volatilization to the atmosphere), provided that the destabilization of the water column occurs. The seasonality of OM production and destruction can be expected to play a significant role in this process. The marine environment has mainly been regarded as a net final sink for POPs, but recent experimental evidence has shown that re-volatilization events from surface water

are more frequent than previously thought, especially in oceanic areas where fast remineralization is expected [22].

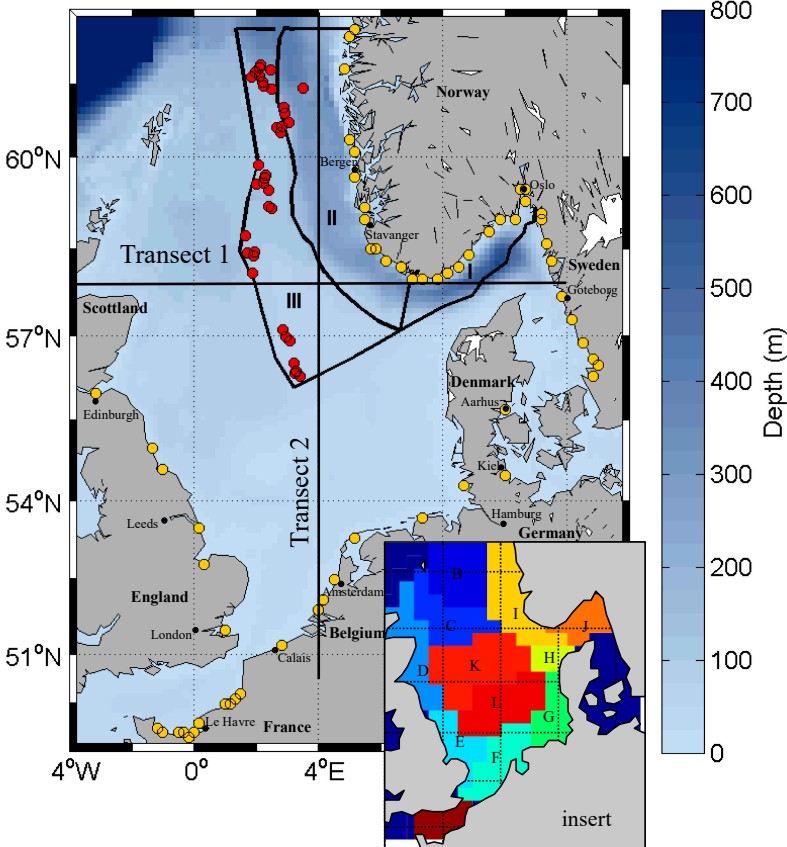

**Figure 1.** Model area and bathymetry, river positions (yellow dots), and Norwegian oil platform positions (red dots). Subdivision of the north-eastern North Sea in areas I–III for time series analysis (see Section 3.2) and transect locations for analysis in Section 3. Insert: area subdivision for model validation (see Section 3.1).

Unfortunately, the scarcity of data on legacy POPs' distribution in the marine water column (the observation data are available mostly for the surface layer) prevents a direct assessment of the process described above. The spatial distribution of the POPs in water columns (horizontal and vertical) is still unknown, thus restricting our understanding of the processes of the POPs' transformation in the environment. Under these conditions, mathematical models of chemical fate and transport are valuable means to theoretically explore the significance of processes. They can facilitate a quantitative understanding, contribute to generating hypotheses, and provide valuable inputs on monitoring design.

To describe the complex physical and biogeochemical drivers that control the fate of chemicals in water, we combined a biogeochemical and a pollutant fate model within a three-dimensional marine hydrodynamic model framework [23]. The goals were (1) to anticipate the structure and the seasonal variability of POP distribution in the water column and to relate it to the seasonality of physical and biological drivers and (2) to theoretically assess to what extend the biological pump-driven export from the photic zone contributes to the permanent confinement of POPs to deep waters. The hydrodynamic framework is coupled with biogeochemical and pollutant-partitioning blocks to simulate the fate and transport of a POP (exemplarily for PCB) in the water column of the North Sea. In contrast to previous similar approaches to model the fate and transport of PCBs in the North Sea [24–26], we considered processes that are related to the seasonality of OM production and destruction that are important for the parameterization of sea water PCB partitioning.

## 2. Methods—Model Description

The model was designed to run in the nested Hamburg Shelf Ocean Model (HAMSOM) model framework [27] and included a pollutant transport module that utilized modelled information for water transport and turbulent diffusion from previously performed long-term simulations [28], a biogeochemical module (Oxygen Depletion Model (OxyDep)) [29], and a pollutant partitioning module (PolPar). In the model framework, pollutants (1) can be transported with advection and turbulent diffusion, (2) get absorbed and released by particulate and dissolved organic matter, and (3) decay. Pollutants enter the model domain through the lateral open boundaries, at the sea surface through atmospheric deposition, and through river loads from land. A permanent loss of pollutants occurs through oceanic transport out of the domain, through decay processes, and through permanent burial to the sediments.

### 2.1. Model Components

To simulate the change of the concentration $C$ of a quantity (e.g., pollutants, oxidized form of inorganic nutrients (NUT), and biota (BIO)) with time t in the ocean grid scale advective transports (second term in Equation (1)), the subgrid scale turbulent diffusion (third term in Equation (1)), biogeochemical/pollutant sink/source terms $R_C$ and sinking need to be considered:

$$\frac{\partial C}{\partial t} + \nabla C \vec{V} - \nabla(K \nabla C) = R_C - \frac{\partial}{\partial z}(w_C C),\qquad(1)$$

where $\vec{V}$ is the current velocity, $K$ is the eddy diffusion coefficient, and $w_C$ is the sinking rate, which is zero for pollutants in the dissolved phase and, in accordance with other studies [30,31], 1 m d$^{-1}$ for the particulate organic material (POM). It was assumed that the chemical pollutant was partitioned between the free fraction ($C_{FREE}$) and fractions that are associated with BIO ($C_{BIO}$), POM ($C_{POM}$) and dissolved organic matter (DOM) ($C_{DOM}$). The fractions of chemical pollutant (PCB) associated with BIO and POM undergo vertical export in association with the sinking material (Figure 2); therefore, they have the same sinking rate as the material they are associated with.

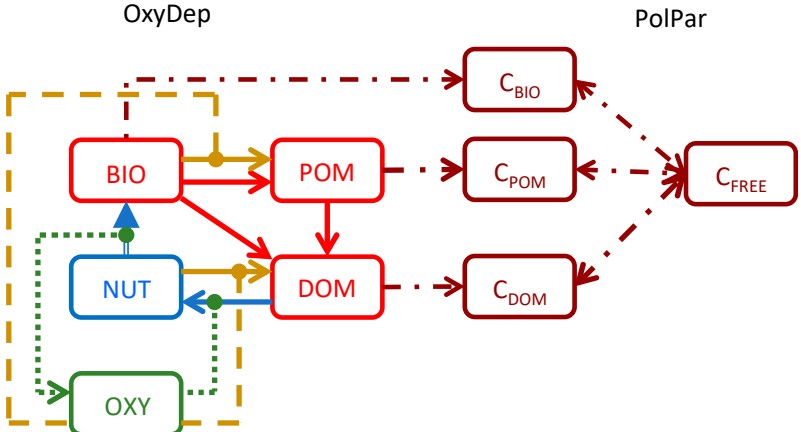

**Figure 2.** Schematic diagram of biogeochemical processes in the biogeochemical module (Oxygen Depletion Model (OxyDep)) and its coupling to the pollutant-partitioning model (PolPar). It is assumed that the total pollutant can be partitioned between pollutant ($C_{BIO}$) in biota (BIO), pollutant ($C_{POM}$) in particulate organic matter (POM), pollutant ($C_{DOM}$) in dissolved organic matter (DOM), and remaining free dissolved fraction ($C_{FREE}$).

### 2.1.1. Modeling Hydrodynamics and Transport with HAMSOM

The hydrodynamic module formed the core (HAMSOM) of the presented model and is part of the ECOSMO (ECOSystem MOdel) model system [23,27]. The model uses a Lax–Wendroff scheme for the

advection equation equipped by a superbee limiter to make the scheme total variation diminishing (TVD); for details, see [32]. As discussed by the latter, this significantly improves the performance of sole upstream schemes and was used by previous POP transport models [24–26]. Turbulent diffusion was implicitly solved by using the eddy diffusivity that was calculated by the product of the eddy viscosity and the Schmidt number (for details, see [33]). To ensure the continuity of the velocity field, we re-diagnosed vertical advection velocity and surface elevation by solving the continuity equation as part of the advection model.

The advection module requires horizontal water transport, turbulent eddy viscosity, and the Schmidt number as the input parameter. Therefore, we used previously calculated daily means from the HAMSOM nested model system. To estimate POP export to the deeper ocean beyond the shelf edge, we combined transport fields from the high resolution prognostic model setup for the coupled North Sea and Baltic Sea (lateral boundary at 60° N) [23] with results from an exterior diagnostic shelf sea model [34]. The horizontal resolution of the advection model was $\Delta \varphi = 6'$ and $\Delta \lambda = 10'$, and in the vertical, the water column was resolved by 20 z-levels with the lower boundaries at 5, 10, 15, 20, 25, 30, 35, 40, 48, 56, 64, 72, 80, 88, 100, 125, 150, 200, 400 and 3500 m. The hydrodynamic model setup has previously underwent extensive validations [35], and it has been shown to well-replicate the interannual variations in the hydrodynamic conditions in the North Sea and Baltic Sea system [36]. From a long term ECOSMO model simulation (1948–2008) [28] that was forced by atmospheric boundary conditions from the NCEP/NCAR re-analysis [37], we selected the years 2004–2008, for which we had atmospheric inputs and high resolution river load data available.

2.1.2. Modeling Biogeochemistry with OxyDep

The aim of the biogeochemical module was to parameterize the seasonality of the production and destruction of organic matter. In detailed ecological models (such as ECOSMO [28]), energy pathways and processes are complex between the single compartment and functional groups, and the net transformation of nutrients is difficult to estimate. Thus, we decided to use a simplified biogeochemical model that allowed for the simulation of the main features of the seasonal transformation of nutrients between inorganic and organic (dissolved and particulate) forms. The idea behind OxyDep is to provide a simple, generalized parameterization of the biogeochemical processes that occur in the water column and in the sediment/water boundary under varying redox conditions. This module was specifically designed to calculate parameters that are used by the chemical fate module.

A schematic diagram of the biogeochemical processes that are considered in the OxyDep is shown in Figure 2. The model was described in detail in [29]. The 5 variables that were considered in the model can be described as follows:

(i) BIO: the concentration of living biota. BIO increases due to primary production, while loss terms are respiration, excretion, and mortality, which include natural mortality rates and intraspecific predation. Though BIO was parameterized to represent both producers and consumers, here it was mainly regarded as a proxy of primary producers:

$$R_{BIO} = \frac{dBIO}{dt} = Growth_{BIO} - Resp_{BIO} - Excr_{BIO} - Mort_{BIO}, \tag{2}$$

(ii) NUT: the concentration of the oxidized forms of nutrients (i.e., $NO_3$ for N):

$$R_{NUT} = \frac{dNUT}{dt} = -Growth_{BIO} + Resp_{BIO} + Decay_{POM} + Decay_{DOM}, \tag{3}$$

(iii) POM: the concentration of all kinds of labile, non-living particulate organic matter. Source terms for POM are the mortality components of BIO, and it declines as a function of autolysis and mineralization:

$$R_{POM} = \frac{dPOM}{dt} = -Auto_{POM} + Mort_{BIO} - Decay_{POM}, \tag{4}$$

(iv) DOM: the concentration of all kind of labile-dissolved organic matter and reduced forms of inorganic nutrients (e.g., $NH_4$ and Urea for N). DOM grows due to excretion and autolysis and declines as a function of mineralization:

$$R_{DOM} = \frac{dDOM}{dt} = Auto_{POM} + Excr_{BIO} - Decay_{DOM}, \tag{5}$$

(v) OXY: the concentrations of dissolved oxygen. OXY changes its concentration following OM production and destruction according to the Redfield ratios.

$$R_{OXY} = \frac{dOXY}{dt} = C_{OtoN}R_{NUT}, \tag{6}$$

OXY is given in μM O, and all other variables are given in μM N. The set of equations used for calculating the biogeochemical components of the model including notations, values, units and the names of the used parameters are given in Appendix A.

### 2.1.3. Hazardous Substances Partitioning and Decay Model: PolPar

PCBs are a group of industrial chemicals that comprise 209 compounds with different chlorination extents [38]. As the focus of our study was to track particularly interesting contamination gradients emerging from the interaction of PCBs with marine organic matter, we considered PCB 153 as a tracer compound. PCB 153 is a highly hydrophobic and persistent PCB with a great potential for bioaccumulation.

Hydrophobic chemicals that are present in the water column engage partitioning between water phase and dissolved (DOM) and particulate organic matter (POM) forms (Figure 2). Partitioning is a function of temperature, compound specific physical–chemical properties, and characteristics of OM [39].

As a first approximation, the partitioning among these phases can be considered instantaneous [40]. This concept is based on the assumption that a high surface/volume ratio of microscale particles (such as phytoplankton) or nano to sub-nano scale dissolved organic matter largely reduces the time to reach diffusive equilibrium.

Here, we assumed that the partitioning of hazardous substances between the dissolved phase and the organic compounds associated with BIO, POM and DOM are governed by the partitioning and mass transfer coefficients that were reported in [40]. The partitioning of hazardous substances with OM is considered to be a fast process, and the shares of the different hazardous substances compartments were calculated with the following system of equations during each time step:

$$
\begin{aligned}
C_{FREE} &= \frac{V_{FREE}(C_{BIO}+C_{POM}+C_{DOM}+C_{FREE})}{(V_{FREE}+K_{BW}V_{BIO}+K_{PW}V_{POM}+K_{DW}V_{DOM})}, \\
C_{BIO} &= \frac{K_{BW}C_{FREE}V_{BIO}}{V_{FREE}} \\
C_{POM} &= \frac{K_{PW}C_{FREE}V_{POM}}{V_{FREE}} \\
C_{DOM} &= \frac{K_{DW}C_{FREE}V_{DOM}}{V_{FREE}}
\end{aligned}
\tag{7}
$$

where $C_{BIO}$, $C_{POM}$, $C_{DOM}$, and $C_{FREE}$ are concentrations of the hazardous substances that are associated with BIO, POM, DOM, and the water-dissolved phase respectively; $V_{BIO}$, $V_{POM}$, and $V_{DOM}$ are volumes that are occupied by BIO, POM, and DOM per water volume ($V_{tot} = 1$ L); $V_{FREE} = V_{tot} - V_{BIO} - V_{POM} - V_{DOM}$; $K_{BW}$, $K_{PW}$, and $K_{DW}$ are the equilibrium partitioning coefficients for the biota/water, POM/water, and DOM/water systems, respectively. This approach for the phase distribution parameterization differs from that used in cases where only particulate organic matter was considered as applied, e.g., in [24,26].

The process of pollutant sorption on organic material in a marine environment depends on many factors such as the organic content and composition of adsorbing matrices, the chemical structure of PCBs, and the lipid fraction in living organisms [39,41,42]. A number of different parameterizations of

$K_{PW}$ and $K_{DW}$ have been reported in the literature both from controlled laboratory studies and field monitoring, with a variance ranging across two orders of magnitude [43]. In addition, no explicit parameterizations are available for $K_{PW}$ and $K_{BW}$ for phytoplankton, so we chose a simplified functional relationship to the known value of $K_{OW}$ ($K_{OW} = 10^{6.9}$): the octanol/water equilibrium partitioning coefficient, as defined in [40]:

$$K_{BW} = K_{PW} = 10 \times K_{DW} = K_{OC} = 0.411 K_{OW}, \tag{8}$$

### 2.1.4. Degradation of the Chemicals

The process of PCB degradation is complicated and varies from one congener to another [41,44,45]. A number of processes simultaneously occur, resulting in a decrease of pollutant concentrations including the influence of sunlight or decomposition due to biological activity [21,39].

Here, the degradation of the chemical in the bulk water phase, $PCB_T$ (as a sum of all PCBs associated with BIO, POM, and DOM and in the water dissolved phase) was parameterized as a first order equation:

$$\frac{\partial PCB_t}{\partial t} = -K_D PCB_t, \tag{9}$$

with the decay coefficients $K_D = 1.3 \times 10^{-4}$ d$^{-1}$ ($1.6 \times 10^{-9}$ s$^{-1}$) calculated based on earlier published rates [21]. To keep generality, this coefficient was calculated as a mean value for total PCBs under consideration of basic processes of bio- and photodegradation.

### 2.2. Model Setup and Boundary Conditions

### 2.2.1. Air–Sea Boundary

Many hydrophobic chemicals including POPs can undergo bi-directional, gaseous air–water exchange. The net direction of the air–sea flux is determined by the chemical activity gradient between the two phases. Marine systems are often regarded as net sinks for airborne POPs, but de-gassing from surface water to the atmosphere is possible provided that high concentrations build up in the water surface. A prerequisite for this process is a perturbation of the equilibrium at the air–water interface. Increasing the concentration of POPs in the surface water under constant (or decreasing) concentrations in the air above can lead to the re-volatilization of some pollutants (such as PCB) to the atmosphere [46]. In open sea areas, where the influence of direct primary emissions to water is less pronounced, re-volatilization may occur, mainly as a result of variability in atmospheric concentrations, resulting, for example, from shifts in air mass direction. In theory, re-volatilization from surface waters may therefore contribute to the depletion of surface water concentrations and alter the vertical distribution of POPs in the water column. Given that the goal of this study was to assess the specific effects of organic matter turn over on the seasonality of POP vertical distribution in the North Sea, we chose to suppress the confounding effects of re-volatilization by imposing that the atmosphere only acts as a source of contaminants for marine waters. For this purpose, monthly information on the total deposition of PCBs for the North Sea was prescribed. The data are available from the European Monitoring and Evaluation Programme (EMEP) (http://www.emep.int) on a 5° × 5° resolution [47].

For the biogeochemical variables, no net mass transfer between ocean and atmosphere was assumed, except for oxygen (OXY), for which the air–sea exchange is given by:

$$Q_{O_2} = k_{660} \left( \frac{Sc}{660} \right)^{-0.5} (Oxsat - O_2), \tag{10}$$

where *Oxsat* is the concentration of oxygen saturation as a function of temperature and salinity according to UNESCO (1986), Sc is the Schmidt number (available from the hydrodynamic model),

and $k_{660}$ is the reference gas-exchange transfer velocity ($Sc = 660$, $CO_2$ at 20 °C). To describe $k_{660}$ as a function of wind speed ($u$), we used the equation proposed by [48]:

$$k_{660} = 0.365u^2 + 0.46u, \tag{11}$$

For estimating the reference gas-exchange transfer velocity in our simulation, we assumed a mean wind speed of 2 m s$^{-1}$.

### 2.2.2. Sediment–Water Boundary

The description of the chemical fate processes that are involved in the water–sediment exchange was simplified, and the sediment was not explicitly solved in the model equation in contrast to the model presented first by [24]. This was a simplification, which was well justified because permanent sedimentation plays only a minor role in large areas of the North Sea [49] due to high re-suspension rates that are caused by strong tidal current in the shallower areas of the North Sea; thus, sediment–water exchange are of minor relevance for the North Sea region.

Here, we considered the seafloor/sediment as a net sink for particulate matter and associated pollutants, and we assumed no re-suspension. Consequently, water phase concentrations decline in the proximity of the seafloor due to burial in the sediments (modified on the base of an approach used in [50]. As described in [29], the sink term $Q_C$ is formulated as:

$$Q_{C_i} = -B_u H_{vert} C_i, \tag{12}$$

where $B_u$ is the burial rate, $H_{vert}$ is the model's vertical resolution, and $C_i$ is the concentration of either BIO or POM.

POPs can be buried together with the particulate OM while, for the scope of this study, partitioning between the bottom water and the sediments was not considered.

The re-suspension of sediments is, in general, expected to play an important role for POP vertical distribution in the North Sea and thus potentially superimpose or counteract the role of the biological pump and OM turn over. Since the latter was the focus of this study, we excluded the re-suspension process from the model formulation and prescribed the sediment–water boundary to act as a permanent sink for waterborne POPs.

### 2.2.3. Lateral Boundaries

The boundary concentrations of living biota and organic dissolved and particulate material were prescribed from the interior by applying Sommerfeld radiation condition [51]. For nitrate (NUT), in contrast, the seasonal changes in vertical distributions were considered and prescribed at the English Channel, North Atlantic, and Kattegat. Data were obtained from the National Ocean Data Centre (NODC; http://www.nodc.noaa.gov/) and the OSPAR (Oslo/Paris convention) homepage (http://www.ospar.org/), and they were used to estimate the typical nutrient distribution for the winter and summer seasons at the boundaries. Riverine NUT and DOM discharges were ignored for simplification. However, other model studies have shown, that, even though river loads clearly impact the magnitude of ecosystem productivity (particularly in near coastal areas), they do not have major impacts on general ecosystem dynamics on the North Sea basin scale [36]. At the open boundaries, we assumed a vertically uniform and constant distributions of chemicals. Concentration values were given as 5 pg L$^{-1}$, according to [52]. The river loads of POPs were taken from the OSPAR database and, for Norwegian rivers, from the Klif database (Klif = klima og forurensningsdirektoratet, i.e., the Norwegian Climate and Pollution department). For the Norwegian rivers, the data are available with monthly resolution, while the load data for other rivers (from Sweden, Denmark, Netherlands, Belgium, Germany, France and UK) are given with annual resolution. For some rivers, data were not available for the full simulation time periods (2004–2006). In these cases, missing data were substituted by means of the existing data for each river. The considered river mouth locations are shown in

Figure 1. At the river mouth locations, the total load was then given as a multiple of the corresponding river discharge.

### 2.2.4. Computational Aspects

The model code was written in FORTRAN-77 and FORTRAN-90 for Linux, and the simulations were realized with a 12 min time step. The simulation started with uniform initial distributions for biogeochemical variables and PCBs. To determine the vertically and horizontally balanced distributions, the calculations were repeated with seasonal irradiance changes and the hydrophysical data (taken for the year 2004) until a quasi-stationary solution of the biogeochemical variables and PCBs was reached.

## 3. Results and Discussion

### 3.1. Model Performance for Biogeochemical State Variables

When compared to the observed magnitude and spatial distribution of nitrogen in the North Sea, we found that the simulated concentrations of nitrogen that were associated with BIO, POM, DOM and NUT (Figure 3) sufficiently represented the observed pattern of phytoplankton, particulate organic nitrogen, dissolved organic nitrogen, and nitrate, respectively, in the North Sea.

Aiming at understanding the seasonal variability of the nitrogen species in the North Sea, Suratman et al. [53] measured the following typical ranges of the surface concentrations of nitrate (0.1–7.2 µM N), ammonium (0.1–2 µM N), DON (the dominant fraction of total nitrogen in all seasons [53], a fact that could not be confirmed by our model) (1.9–11.2 µM N), and PON (0.3–5.6 µM N) in the western central part of the North Sea. Our estimates for this region (Figure 3) were clearly in the same range of magnitude and furthermore represented the observed seasonality with high NUT concentrations in winter (Figure 3b) and nitrate depletion in the surface waters in summer (Figure 3a). Additionally, DOM concentrations were slightly underestimated by the model. The discrepancies might have stemmed from the fact that the DOM model only represented labile autochthonous organic nitrogen (and ammonia), while the observed one also included more stable allochthonous organic matter supplied by rivers and the currents. Chlorophyll-a (Chl-a) in the North Sea was found to vary between 0.1 and 6.2 µg L$^{-1}$ [53], which was approximately 0.03–1.6 µM N. The model simulated BIO showed a comparable range (Figure 3).

The spatial distribution of nitrate was measured by [54] in October–November 1998. The modelled spatial distribution of NUT in winter (Figure 3b) showed remarkable similarities to the observed pattern. Both the observed and modelled patterns featured a concentration maximum in the shallow southern and south-eastern parts of the North Sea as well as along the British Coast and in the Atlantic water in the north-western North Sea. Local minima with less than 2 µM were modelled and observed in the central North Sea, above Dogger Bank, and in the north-eastern part off the Norwegian coast.

For a more quantitative validation of the modelled surface nutrient concentration, we used observed surface nitrate data from the ICES (International Council for the Exploration of the Sea) database (www.ices.dk) and co-located model estimates with observations in time and space for the year 2006. Subsequently, we estimated seasonal cycles and relevant statistics presented in a Taylor diagram [55] (Figure 4). In Figure 4, we show only estimates for regions (regions are shown in the insert of Figure 1) where enough data were available for a comparison of the seasonal cycle. The analysis indicated that the model was able to represent observed seasonal dynamics in surface nutrients in all of the regions. Especially good correlations were estimated for the Norwegian trench regions (Figure 4(bI,bJ)) and for the central North Sea (Figure 4(bL)). Though the model realized the seasonality of nitrate in the coastal regions (Figure 4(bF–bH)), it apparently underestimated nutrient concentration in winter and early spring. This discrepancy indicated that the model's capability to simulate near coastal ecosystem dynamics was restricted by the missing nutrient loads from land. When compared to result from the fully coupled biophysical model ECOSMO II [28], we found the spatial pattern in surface biomass (phytoplankton and zooplankton) well-resembled the presented results for BIO.

ECOSMO calculated concentrations in surface biomass ranging from 0.7 to ~5. µM N, which was locally
(especially along the coast) higher than the estimates from OxyDep (Figure 3). This deviance once more
could be associated with the missing river nutrient loads. Despite its simplifications, the comparison
to observations and to the validated model ECOSMO II verified the model approach of OxyDep and
allowed for the conclusion that OxyDep estimates are useful for calculating pollutants' fates in the
North Sea at seasonal and inter-annual time scales.

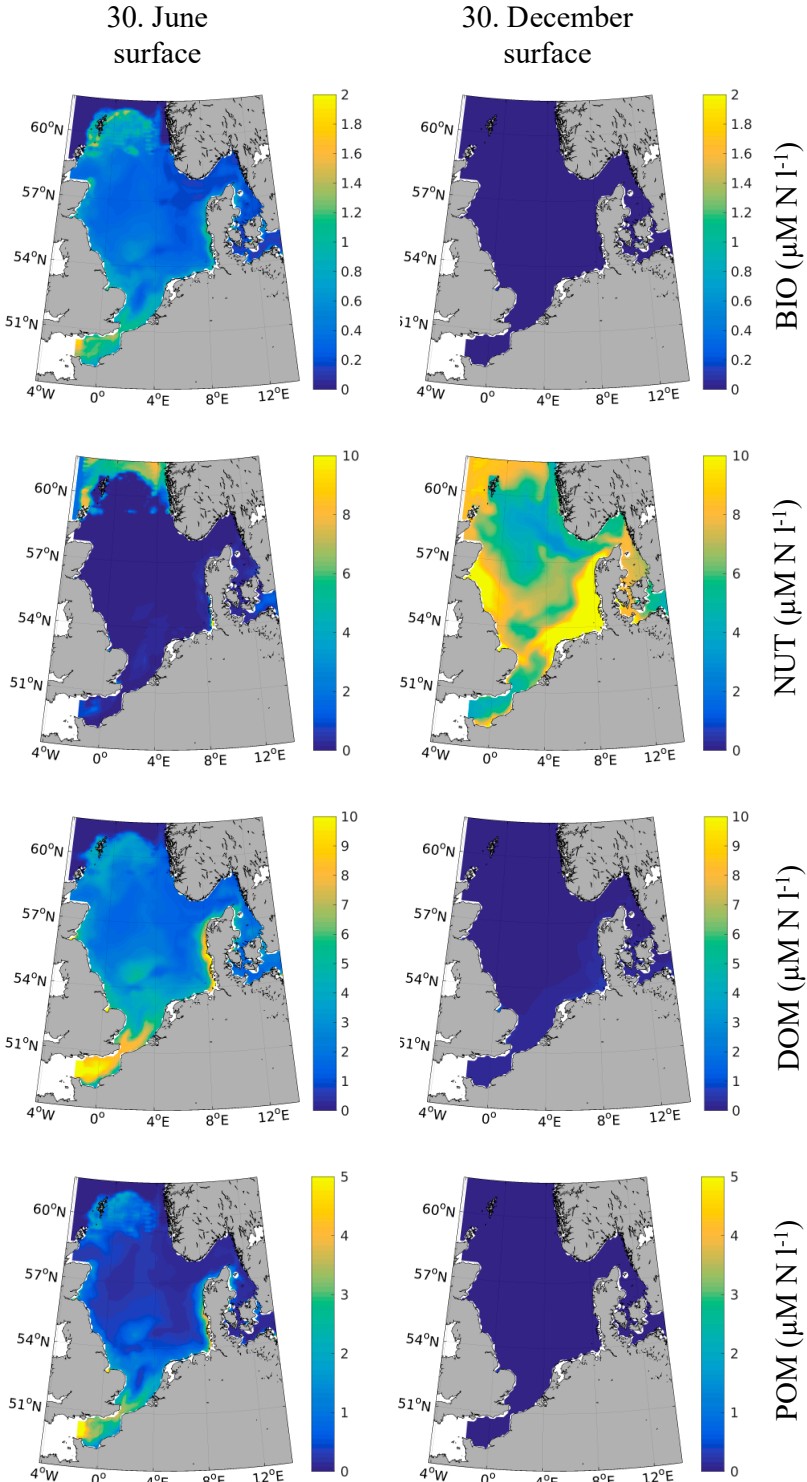

**Figure 3.** Surface distribution of the variables BIO, DOM, NUT (representing nitrate in the model),
POM in June (**left**) and December (**right**) as estimated in the biogeochemical module OxyDep.

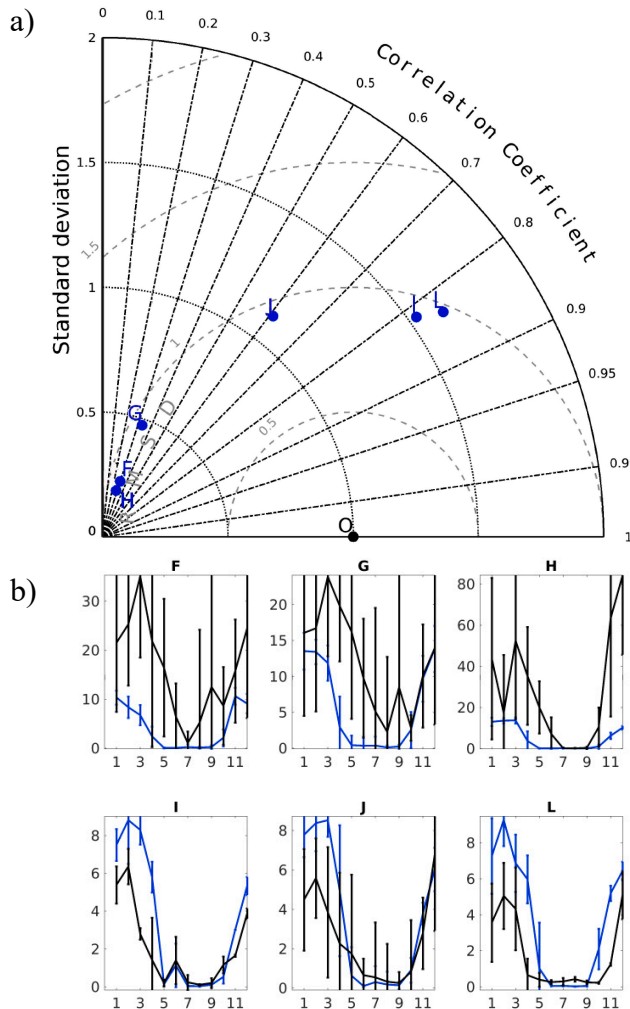

**Figure 4.** Validation of 2006 surface nitrate (model (blue) versus ICES data (black) (www.ices.dk)). Taylor diagram (**a**) and seasonal cycle (**b**) in different sub areas of the North Sea. Area separation according to Figure 1 insert.

The vertical distributions of the biogeochemical parameters along longitudinal transects (Figure 5) demonstrated that during summer, a subsurface maximum of BIO was formed as a result of stratification and near surface nutrients depletion. Subsurface production occurs in a depth interval between 10 and 25 m, as has been described earlier in, e.g., [56], and which is in accordance to observations [57].

*3.2. Spatial-Temporal Variability in PCB*

In contrast to the pioneer work of [24], our model considered DOM as an explicit model compartment: Partitioning with dissolved organic matter represented a significant pool of the total pollutant, and this affected the estimates of the gravitational flux of the chemical pollutants bound with particles (POM). According to experimental data, the sorption of lipophilic congeners to DOC could have resulted in the underestimation of partition coefficient values by a factor of five-to-seven [58]. The partitioning properties of DOC can vary widely depending on DOC composition. In the model, the considered DOM represented the "fresh" DOC originated in sea water. Therefore, it consisted of more compounds of low molecular weight (LMW), which made it more labile with respect to microbial activity [59] when compared to the DOC transported from the rivers.

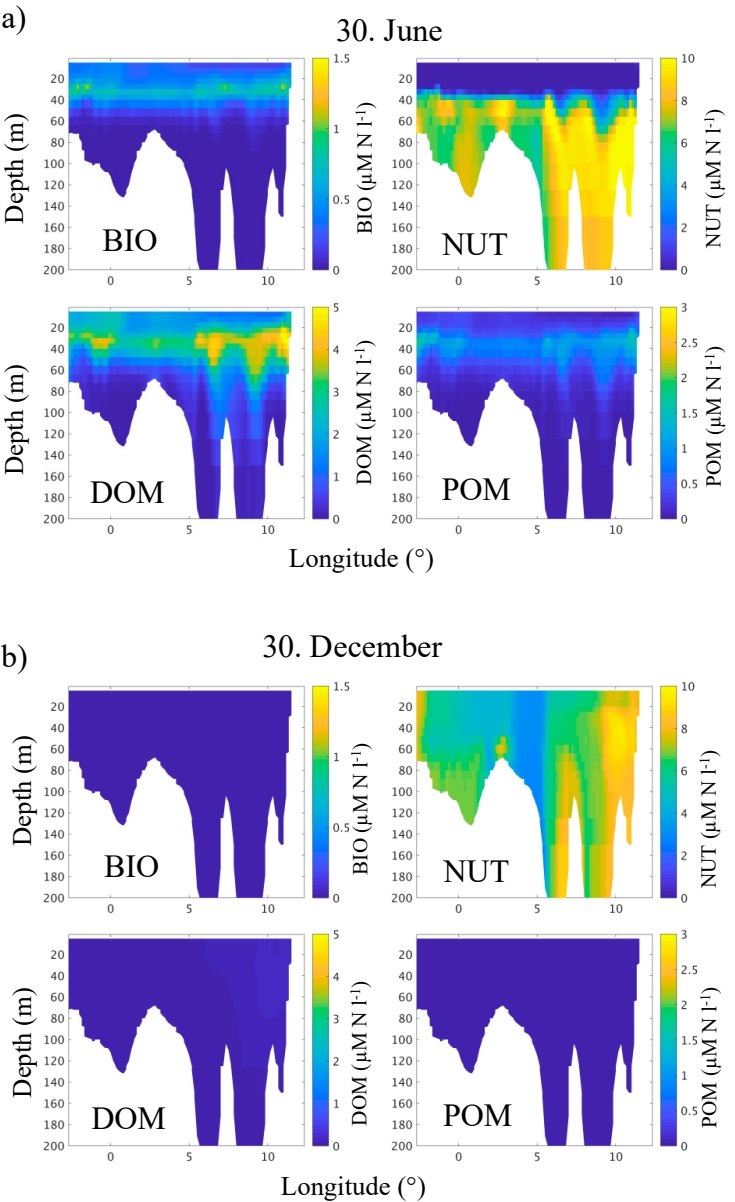

**Figure 5.** Vertical distribution along transect 1 (58° N see Figure 1) of the variables BIO, DOM, NUT, POM in June (**a**) and December (**b**), as estimated in the biogeochemical module OxyDep.

The simulated horizontal and vertical distribution of total PCB (particulate and dissolved, i.e., a sum of $C_{BIO}$, $C_{POM}$, $C_{DOM}$, and $C_{FREE}$) and particulate PCB (a sum of $C_{BIO}$ and $C_{POM}$) are shown in Figures 6 and 7. The calculated concentrations of the total PCB$_t$ were in the limits of several pg L$^{-1}$ up to several hundreds of pg L$^{-1}$, which corresponded to observational data ranges [52] and were several orders of magnitude smaller than the EEC (European Economic Community) maximum permissible level of PCB for drinking water (0.5 µg L$^{-1}$, [60]). In [61], environmental assessment criteria (EACs) for seven PCB congeners were listed for water, sediments, and higher trophic level groups. EACs are set to define the level of pollution below which the pollutant should not cause chronic effects in sensitive marine species and should thus present no significant risk to the environment. For the PCBs listed in the report, the EAC in water was in the range of 0.02–1 ng L$^{-1}$. It is important to highlight that safety thresholds for chronic effects of PCBs have been widely debated, reaching little agreement among the scientific community. At these low concentrations in the abiotic environment, PCBs have a relevant toxic activity for organisms at the higher trophic levels where they are up-concentrated by several orders of magnitude through the biomagnification process. It is well documented that, even at

considerably lower water concentrations (such as in the Arctic) PCBs can be at the base of adverse chronic effects in top predators [62]. As the near-coast regions, which show highest concentrations in the North Sea, are not only heavily fished but can also be considered the major nursery grounds for a multitude of marine fish species in the North Sea [63], even low concentrations of PCB might potentially affect their marine ecosystems.

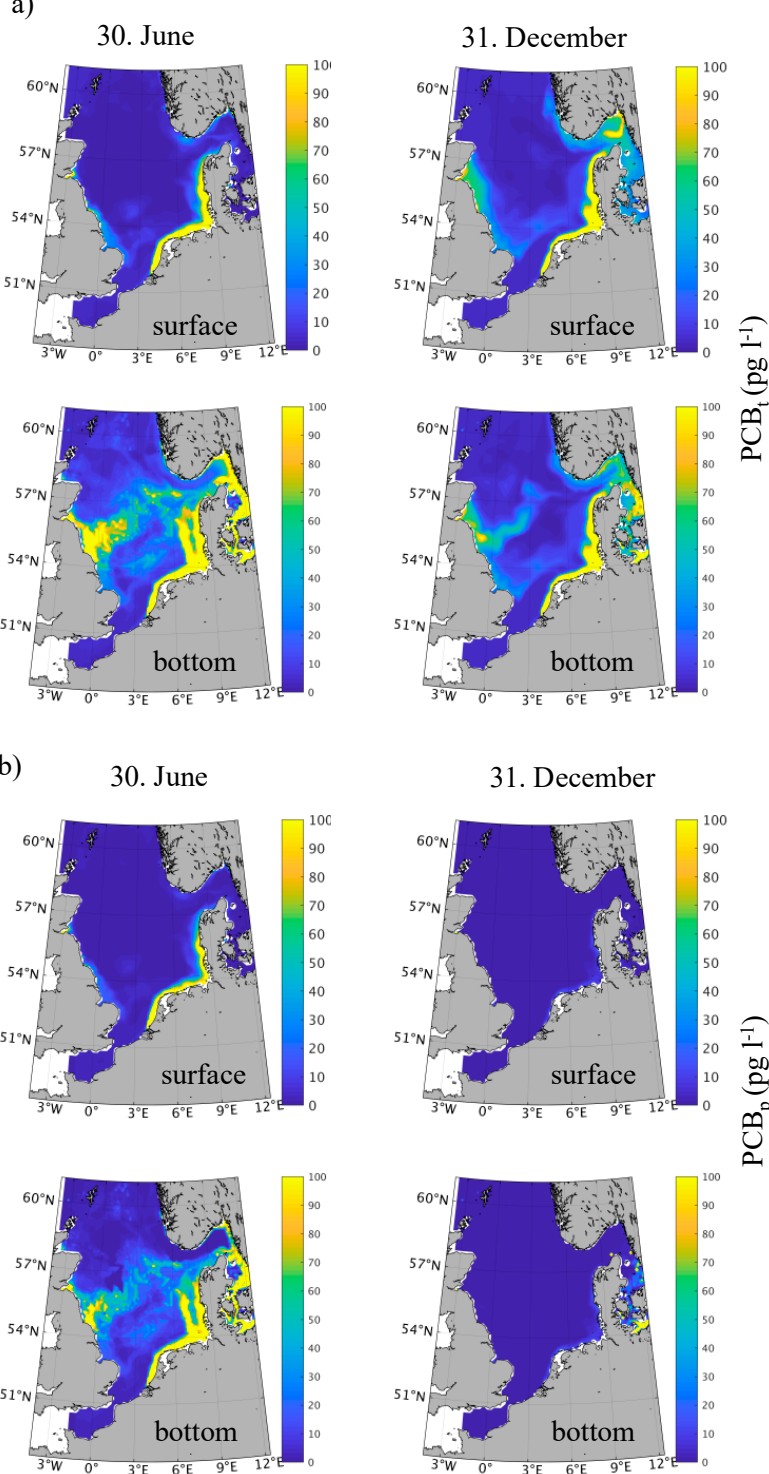

**Figure 6.** Distributions of total polychlorinated biphenyls (PCB$_t$) (**a**) and particulate PCBs (PCB$_p$) (**b**) at the surface and at the bottom in June and December 2006.

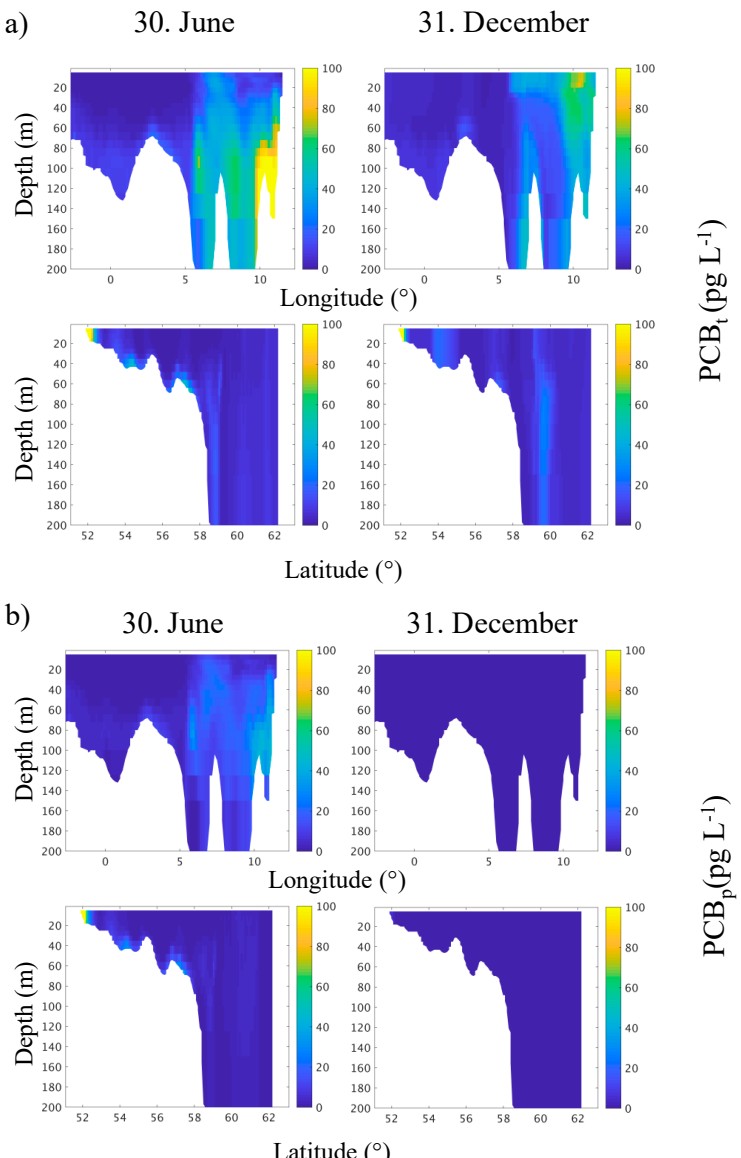

**Figure 7.** Vertical distributions of simulated total PCB ($PCB_t$) (**a**), and particulate PCB ($PCB_p$) (**b**) along transect 1 (upper panels) and transect 2 (lower panels) in June (left) and December (right).

Two aspects of the spatial distribution of surface PCBs are the anthropogenic PCB production and the associated discharges from land [24,26,64–68]. Though their production is currently banned by the Stockholm Convention (http://www.pops.int/access 09.12.2019), these contaminants are still entering the environment via leaching and weathering from already existing products. Accordingly, the calculated distributions of PCBs in the surface layer (Figure 6) were found to feature the highest pollution, with concentrations exceeding 100 pg/l, along the southern coast of the North Sea and along the coastline of Germany and Denmark. According to [25], the observed concentrations of total concentrations of PCB 153 in the German Bight were in the range of 0.003–0.26 ng $L^{-1}$ in the time period from 1995 to 2002, but they had a slight increasing trend to the end of that period, with maximum values along the coast and especially close to the Elbe estuary. This pattern was consistent with our model results both in magnitude and spatial distribution. Additionally, local maxima in PCB concentrations were estimated along the Norwegian Coast in the vicinity of the Oslo Fjord, Brevik in the Grenlandsfjord area, and Haugesund. In the offshore areas and the central North Sea, in contrast, were surface concentrations below a few pg $L^{-1}$. Our predictions were also consistent with results from [69] that showed concentrations of PCBs (sum of seven major congeners) typically ranging

between less than 1 and 20 pg L$^{-1}$, with the higher levels occurring in the proximity of the German Bight and major river estuaries.

According to the model (Figure 6a,b), the fraction of PCBs accumulated in particulate matter (including living organisms and detritus) dominated the total PCB content in the productive and polluted regions of the North Sea (i.e., along the Danish coast) during the spring and summer period, which can be confirmed by observations [70].

The model demonstrates a seasonal signal in the vertical distribution of PCBs in the water column (Figure 7). The synthesis of OM during spring and summer production resulted in the consumption of the dissolved pollutant into its particulate form due to partitioning processes (Figure 8). In contrast to the model described by [24,26], a separation between particulate and dissolved fractions of pollutant led to a simulated maximum of the total pollutants at an intermediate depth between 50 to 100 m. This intermediate maximum was especially pronounced during the stratified summer period, while thermal convection due to autumn and winter cooling resulted in a homogenization of the water column in winter and the destruction of the subsurface maximum of PCBs in the process. However, the vertical maximum of PCBs in the dissolved phase was maintained in the stratified Norwegian Coastal zone and the Skagerrak. The sinking of pollutant-enriched organic particles resulted in a decrease of the total pollutant concentrations in the euphotic zone (from the surface down to 30–40 m) and an increase of concentrations in the intermediate waters below. The latter coincided with the bottom layer in the shallower areas. Here, the pollutants were returned to their dissolved form in the process of decay of particulate and dissolved organic matter, resulting in a vertical pollutant maximum. During winter, biological production was almost absent, and the water column was mixed by thermal convection homogenizing the pollutant concentration (Figure 7). A significant fraction of the pollutants, exported during the productive season, was then returned to the surface. This was illustrated by elevated surface PCB concentrations in December compared to June (Figure 8). Thus, the model results verified the "biological pump" as a key mechanism for the formation vertical stratification in pollutant concentrations and thereby confirmed earlier theoretical hypotheses by [11,12,14].

The modelling results demonstrated that the local enrichment of the bottom layer with pollutants (here a PCB) (Figure 7). These pollution hot spots change concentration and position with time, and they might be affected by a combination of hydrodynamical processes (i.e., meanders and eddies), seasonal scale biogeochemical processes, and interannual, seasonal, and sporadic changes in the supply of pollutants with the rivers and the atmosphere. In addition the timing of the pollution hot spots is impacted by changes in the sinking characteristics of particulate organic matter. It has been shown that the sinking velocity on POM changes seasonally depending on the source of the material [71]. This process was not considered in our model formulation, which led to a change in the timing of the occurrence of the described hot spots. However, the general underlying process of pollution export to the ocean's interior and the associated conclusions remained intact. These distribution changes were in accordance to findings by [26].

To estimate the seasonal variability and interannual variability, we integrated the total PCB concentrations for three regions (regions I, II, III), positioned in the Norwegian zone of the Sea, as shown in the map Figure 1. The region -averaged total PCB concentrations (Figure 9) were, on average, higher in region I (36–55 pg L$^{-1}$) compared to region II (11–27 pg L$^{-1}$) and region III (2–7 pg L$^{-1}$), which is a more offshore region The time series for both the total PCB and particulate PCB concentration indicated interannual variation up to 50%–100% of an averaged value in all three regions. Only a little seasonality was estimated for the total (dissolved and particulate) PCB, especially in region I and III, while, in contrast, particulate PCB was characterized by a clear seasonal variability with a maximum in spring and a minimum in winter (see also Figure 8). This was associated with the seasonality of the biological activity, as the seasonality in the processes that are related to dissolved and particulate PCB are principally anticorrelated. Therefore, variations in region-averaged concentrations were mainly related to interannual and seasonal changes in PBC sources such as atmospheric fluxes and river loads, as well as the interannual variation in circulation. Since regions I and II are especially

impacted by the Baltic Sea and the boundary to the North Atlantic, the latter would lead to variations in export of pollutants to the North Atlantic and variations in pollutant exchange with the Baltic Sea. Since interannual variations in river and platform loads were not considered in the model setup, they could not be used to explain these area specific differences.

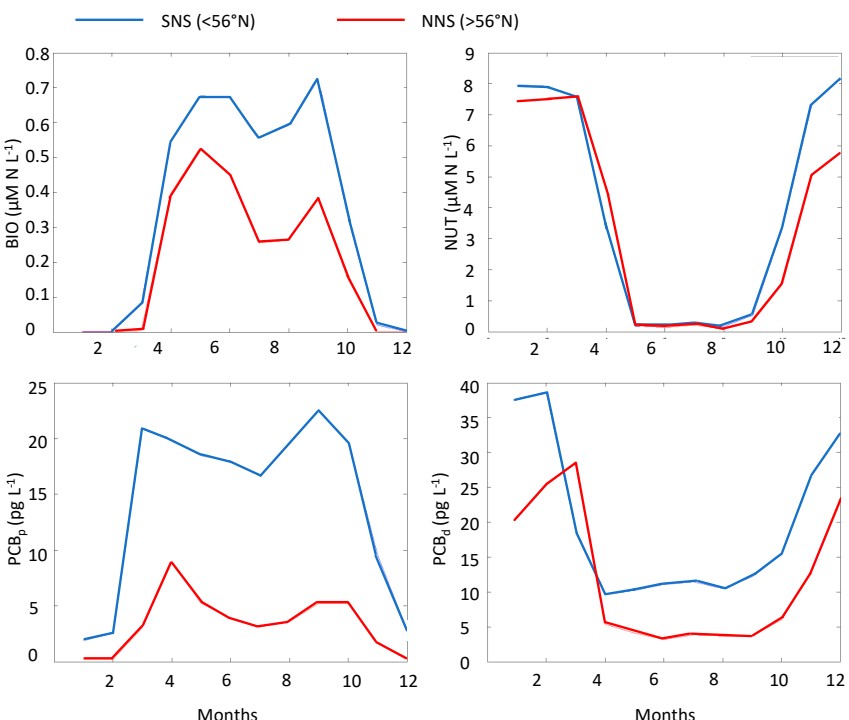

**Figure 8.** Estimated mean seasonal cycle of surface BIO, NUT, dissolved PCB ($PCB_d$), and particulate PCB ($PCB_p$). Average monthly mean values are given for the northern North Sea (NNS > 56° N) and for the southern North Sea (SNS < 56° N).

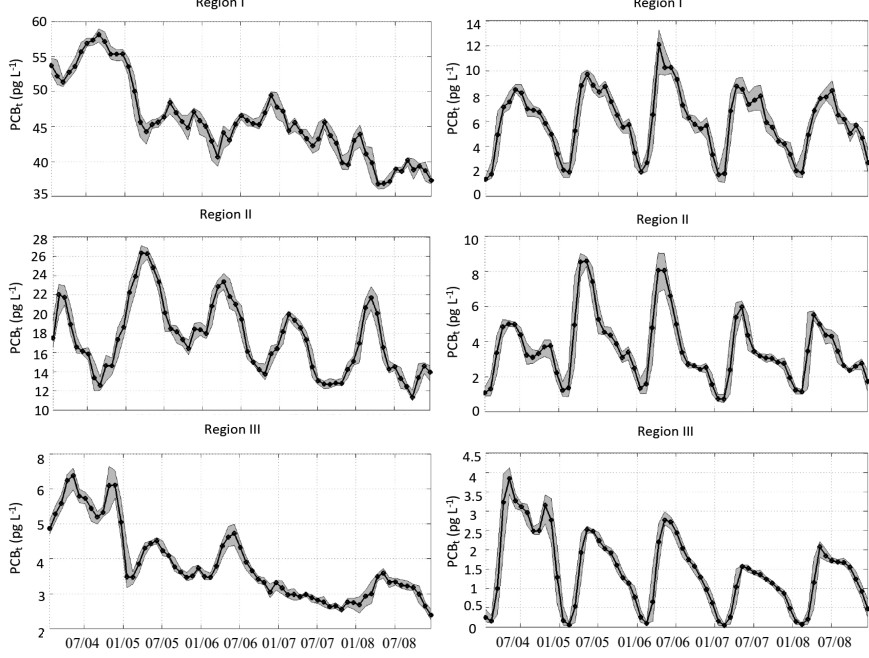

**Figure 9.** Temporal variability of the volume-averaged concentrations of total $PCB_t$ (**left**) and particulate $PCB_p$ (**right**) in the regions I (**top**), II (**middle**), and III (**bottom**) in 2004–2008 (for region separation, see Figure 1). (Note that the *y*-axis scales vary in magnitude).

## 4. Conclusions

The here-described model continues the work on the three-dimensional modelling of the distributions and variability of hazardous substances in the seawater. The model took transport by currents, net sedimentation, partitioning with OM, degradation, air–sea exchange, transport with rivers, and burial into the sediments into consideration. In contrast to the previously proposed models [24,26], our model considered separate compartments for dissolved and particulate organic matter, thus allowing for a process description of partitioning pollutants with living organisms, detritus, and dissolved organic matter. The model was able to reproduce the spatial distribution of PCB in the North Sea and to simulate its temporal variability.

The major results of this study can be summarized as follows:

The model results suggest that a vertical stratification of PCB must form in the water column, with a minimum in the surface layer (better pronounced in summer) and a maximum in the intermediate water (in shallow waters near the bottom) as a consequence of biological production in the surface and matter degradation in the subsurface. Thus, the vertical concentration can vary by factors up to several hundreds.

Hot spots in pollutant concentrations specifically form in high productive regions, such as Fisher Banks, that are highly relevant for the ecosystem and fisheries. This implies that those regions, which are particularly relevant as feeding grounds for marine seafood, are highly polluted, even though the average concentrations and surface concentrations might indicate a good environmental status of the marine waters. These heavily loaded regions can lead to environmental stress for marine fish and amplify toxic stresses and bioaccumulation, which could finally lead to highly contaminated seafood even in unpolluted marine waters, as found in, e.g., different areas of the North Sea and Norwegian Sea by [71–73].

The river loads of pollutants play a significant role in the formation of horizontal spatial variability, with maxima near the coastal discharges from the industrial regions. These areas show elevated concentrations several hundred times larger than surface offshore concentrations. Additionally, river loads in the North Sea affect larger areas along the shore, as they are transported with the prevailing, predominantly cyclonic circulation pattern and thus lead to generally higher concentrations along the coast.

Our results have various implications. (i) When modelling the fate and transport of pollutants in the atmosphere, the ocean has to be considered as a spatially and temporally varying potential secondary source for pollutants. (ii) To resolve the fate and transport of pollutants in the ocean, it is necessary to resolve the biological pump in space and time. Hence, the complementation of the physical transport model by a biogeochemical module that resolves both particulate and dissolved organic compartments is imperative.

Beyond the scientific aspects, our results have consequences for marine environmental and risk assessments, such as those carried out as a follow up of the Marine Strategy Framework Directive (MFSD, European Parliament and the Council, 2008). Environmental assessments need to go beyond the current focus on average and surface concentrations, and they should include the identification of hot spots in marine waters as well as a potential exposure risks assessment for key marine species.

**Author Contributions:** Conceptualization, C.S., E.V.Y., L.N.; methodology, U.D., C.S., E.V.Y.; software, U.D., C.S., E.V.Y.; validation, U.D., C.S.; visualization, U.D., E.M.; resources, C.S.; writing—original draft preparation, U.D., E.V.Y., C.S.; writing—review and editing, U.D., E.Y., C.S., L.N., E.M. All authors have read and agree to the published version of the manuscript.

**Funding:** CS and EY received funding from the Klima- og Foruensingsdirektoratet, Norway (Tilførselsprogrammet 2010). EY was supported by Norwegian Research Council project no. 272749 ('Aquatic Modeling Tools', SkatteFUNN) and by the Ministry of Science and Education of Russia (theme No. 0149-2019-0003). EM received funding from the H2020 project "Strengthening the European Research Area in the domain of Earth Observation" grant agreement number 689443. This work is a contribution to the EXC 2037 'Climate, Climatic Change, and Society'—Project Number: 390683824 funded by the Deutsche Forschungsgemeinschaft (DFG, German Research Foundation).

**Acknowledgments:** The authors thank our colleagues K.L. Daae, K. Borgå, N. Green, and David Drewes for help with data preparation, figures and discussions. We would also like to thank three anonymous reviewers for their helpful comments on an earlier version of the manuscript.

**Conflicts of Interest:** The authors declare no conflict of interest.

## Appendix A

The set of equations that were used to simulate sink and source terms for the state variables (BIO, NUT, POM, DOM, and OXY) in OxyDep is given in the following. In the model, only the variable for oxygen OXY is given in $\mu$M O, while all other variables are calculated in $\mu$M N. Notations, values, units, and names of parameters used are given in Table A1.

BIO: The specific growth rate of BIO is a function of temperature (T, °C), light, and nutrient availability with the maximum specific growth rate $K_{NF}$:

$$Growth_{BIO} = K_{NF}f(T)f(i)f(NUT)BIO,$$

$$f(T) = \frac{0.2 + 0.22\left(e^{0.21T} - 1\right)}{\left(1 + 0.28e^{0.21T}\right)},$$

$$f(I) = f(\varphi)\frac{I}{I_{opt}}e^{\left(1 - \frac{I}{I_{opt}}\right)}; \quad I = I_0 e^{-kh}; \quad h: \ depth,$$

$$f(\varphi) = \cos\left(\varphi - 23.5\sin\left(\frac{2t}{365.2}\right)\right); \quad t: \ time \ (Julian \ days);$$

$$f(NUT) = \frac{(NUT/BIO)^2}{(NUT/BIO)^2 + K_{NUT}},$$

The excretion of BIO ($Excr_{BIO}$) is calculated as:

$$Excr_{BIO} = K_{BD}BIO,$$

The mortality rate of BIO ($Mort_{BIO}$) is parameterized by using specific rates for mortality $K_{BP}$ in oxic and $K_{BP}{}^A$ in anoxic conditions:

$$Mort_{BIO} = K_{BP}BIO + f_S(OXY)K_{BP}^A BIO + K_{BP}^C \ (0.5 \ (1 - \tanh(BIO_{Can} - K_{Can}BIO)BIO,$$

The last term in the equation describes the additional mortality due to "cannibalism," which starts when the BIO exceeds the threshold value $BIO_{Can}$.

POM/DOM: We then considered the formation of DOM from POM (autolysis) ($Diss_{POM}$) with a constant specific rate $K_{PD}$:

$$Diss_{POM} = K_{PD}POM,$$

The DOM and POM decay takes place due to oxic decay (first term in the following equation) and denitrification under suboxic conditions (second term):

$$Decay_{DOM} = K_{DOM}f_t^D(T)DOM + K_{DOM}^S f_t^D(T)f_S(OXY)f_N^D(NUT)DOM,$$

where $f_t^D(T)$, $f_N^D(NUT)$, and $f_S(OXY)$ are dependences of $Decay_{DOM}$ on temperature, NUT, and OXY, respectively.

The POM decay ($Decay_{POM}$) was parameterized as:

$$Decay_{POM} = K_{POM}f_t^D(T)f_0(OXY)POM + K_{POM}^S f_t^D(T)f_S(OXY)f_N^D(NUT)POM,$$

with,

$$f_t^D(T) = B_{da}\frac{T^2}{T^2 + T_{da}{}^2},$$

$$f_N^D(NUT) = (1 - \tan h(NUT_{Den} - NUT),$$

The changes between the processes occurring in oxic and suboxic conditions were parameterized with soft switches based on hyperbolic tangent functions.

$$f_O(OXY) = 1 - 0.5(1 + \tan h\left(OXY - O_2^{bf}\right),$$

$$f_S(OXY) = 0.5(1 + \tanh\left(OXY - O_2^{bf}\right)),$$

Changes in the OXY content were calculated by using the Redfield ratio $C_{OtoN}$.

**Table A1.** Notations, values, units and names of parameters used in OxyDep.

| Notation | Value | Units | Parameter |
|---|---|---|---|
| *GrowthBIO* | | d$^{-1}$ | Specific growth rate |
| $f(I)$ | | - | Photosynthesis dependense on irradiance |
| $f(T)$ | | - | Photosynthesis dependense on temperature |
| $f(\varphi)$ | | - | Variation of light with latitude and time |
| $f(NUT)$ | | - | Photosynthesis dependense on nutrient |
| $K_{NF}$ | 4.0 | d$^{-1}$ | Maximum specific growth rate |
| $I_0$ | 80. | W m$^{-2}$ | Optimal Irradiance at the surface |
| $k$ | 0.10 | m$^{-1}$ | Extinction coefficient |
| $I_{opt}$ | 25. | W m$^{-2}$ | Optimal irradiance |
| $\varphi$ | | ° | Latitude |
| $b_m$ | 0.12 | °C$^{-1}$ | Coefficient for uptake rate dependence on t |
| $c_m$ | 1.4 | - | Coefficient for uptake rate dependence on t |
| $K_{NUT}$ | 0.02 | | Half-saturatuib constand for uptake of NUT by BIO (squared Mikhaelis-Menten dependence) |
| r_bio_nut | 0.05 | d$^{-1}$ | Specific respiration rate |
| $K_{BP}$ | 0.01 | d$^{-1}$ | Specific rate of mortality |
| $K_{BD}$ | 0.10 | d$^{-1}$ | Specific rate of excretion a |
| $K_{BP}{}^A$ | 0.5 | d$^{-1}$ | Specific rate mortality in anoxic cond. |
| $K_{BP}{}^C$ | 0.6 | d$^{-1}$ | Spec.rate of addtional mortality (cannibalism) |
| $BIO_{Can}$ | 1. | μM N | Threshold BIO value for cannibalism |
| $K_{PD}$ | 0.10 | d$^{-1}$ | Specific rate of POM decomposition (autolis) |
| Decay$_{DOM}$ | | d$^{-1}$ | mineralization of POM |
| $K_{POM}$ | 0.003 | d$^{-1}$ | Specific rate of POM oxic decay |
| $K_{POM}{}^S$ | 0.001 | d$^{-1}$ | Specific rate of POM denitrification |
| Decay$_{DOM}$ | | d$^{-1}$ | mineralization of DOM |
| $K_{DOM}$ | 0.05 | d$^{-1}$ | Specific rate of DOM oxic decay |
| $K_{DOM}{}^S$ | 0.0005 | d$^{-1}$ | Specific rate of DOM denitrification |
| $T_{da}$ | 13. | - | Coefficient for dependence of decay on t |
| $B_{da}$ | 20. | - | Coefficient for dependence of decay on t |
| $B_u$ | 0.22 | d$^{-1}$ m$^{-1}$ | Burial coeficient for lower boundary |
| $NUT_{Den}$ | 1. | μM N | Threshold NUT value for denitrification |
| $O_2^{bf}$ | 20. | μM | Constant that defines the oxygen threshold |
| $C_{OtoN}$ | −8625 | - | O to N Redfield ratio (138/16) |

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
