# Peer review of "Understanding the Role of Organic Matter Cycling for the Spatio-Temporal Structure of PCBs in the North Sea"

_water, doi:10.3390/w12030817_

Round 1
Reviewer 1 Report
Cooments are attached in document

Reviewer 2 Report
I appreciated to read this well done paper, of interest for the science readers. I had difficult to found problems or criticisms.
Well Done!!
Author Response
We would like to thank the reviewer for taking the time and evaluate our manuscript.
Round 2
Reviewer 1 Report
Regarding figure 9, I still think you will be better of if you put a grid that demonstrates the different Y axis. also the X axis labels could be rotated and font size increased
Author Response
Regarding figure 9, I still think you will be better of if you put a grid that demonstrates the different Y axis. also the X axis labels could be rotated and font size increased
response: We agree to change the x-axis label as requested. However, we don't quite understand what the reviewer means by "put a grid that demonstrates the different Y axis". We would rather keep the y-axes as they are because the magnitudes are very different . With standardized y-axis it would be dificult to visualize the variations appropriately.